# ToVE: Efficient Vision-Language Learning via Knowledge Transfer from Vision Experts

**Yuanchen Wu**[1,2,*]**, Junlong Du**[2]**, Ke Yan**[2†]**, Shouhong Ding**[2]**, and Xiaoqiang Li**[1†]

[1]School of Computer Engineering and Science, Shanghai University, Shanghai
[2]Tencent Youtu Lab, Shanghai
{yuanchenwu,xqli}@shu.edu.cn, {jeffdu,kerwinyan,ericshding}@tencent.com

## Abstract

Vision-language (VL) learning requires extensive visual perception capabilities, such as fine-grained object recognition and spatial perception. Recent works typically rely on training huge models on massive datasets to develop these capabilities. As a more efficient alternative, this paper proposes a new framework that **T**ransfers the knowledge from a hub **o**f **V**ision **E**xperts (**ToVE**) for efficient VL learning, leveraging pre-trained vision expert models to promote visual perception capability. Specifically, building on a frozen CLIP encoder that provides vision tokens for image-conditioned language generation, ToVE introduces a hub of multiple vision experts and a token-aware gating network that dynamically routes expert knowledge to vision tokens. In the transfer phase, we propose a "residual knowledge transfer" strategy, which not only preserves the generalizability of the vision tokens but also allows detachment of low-contributing experts to improve inference efficiency. Further, we explore to merge these expert knowledge to a single CLIP encoder, creating a knowledge-merged CLIP that produces more informative vision tokens without expert inference during deployment. Experiment results across various VL tasks demonstrate that the proposed ToVE achieves competitive performance with two orders of magnitude fewer training data.

## 1 Introduction

The integration of visual perception with language processing, referred to as vision-language (VL) learning, is a critical frontier in multi-modal research. Compared to standalone language processing, it is a comprehensive super-set that necessitates additional visual perception capability. Many VL tasks, such as image captioning (Lin et al., 2014) and visual question answering (VQA) (Antol et al., 2015), require the model to be capable of content understanding, fine-grained recognition, and spatial perception. Recent works have predominantly relied on massive datasets (sometimes over **billions** of image-text pairs) with large-scale model architectures (Wang et al., 2022a; Li et al., 2023a; Wang et al., 2023) to develop these capabilities from scratch. However, the dependency on a massive dataset presents significant challenges, particularly in specialized domains such as medical imaging where acquiring vast amounts of data is not feasible.

To achieve efficient VL learning, one direct approach is to train from scratch using a small-scale dataset and model architectures. However, the overall visual perceptual capabilities of these models exhibits a significant degradation due to insufficient learning of diverse visual perceptual skills. Although some studies (Dai et al., 2022; Liu et al., 2024b) have attempted to address this issue by transferring the image-text pre-trained CLIP (Radford et al., 2021) to VL tasks, recent findings indicate that CLIP's visual perception capability is also limited (Li et al., 2022a; Tong et al., 2024). As illustrated in Figure 1, the recent advanced efficient Vision-Language Model (VLM) equipped with CLIP, Prismer-Z (Liu et al., 2024b), ***struggles with spatial reasoning and fine-grained perception***, often misinterpreting spatial relationships and failing to differentiate between visually distinct objects. Moreover, in tasks such as image captioning, this model ***is prone to visual hallucinations***,

---

*This work was done during an internship at Tencent Youtu Lab.
†Correspondence to: Ke Yan and Xiaoqiang Li.

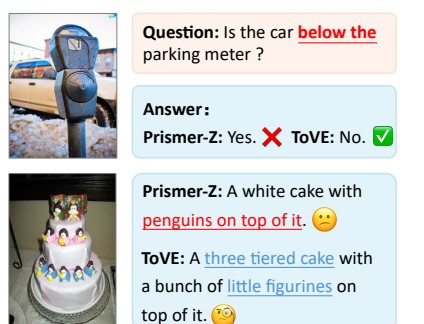

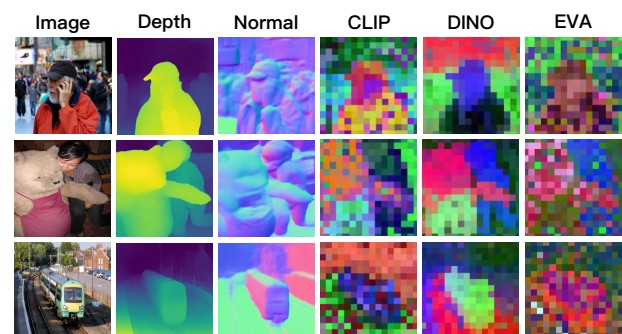

Figure 1: **The *comparison* between Prismer-Z and the proposed ToVE** on Novel Object Caption and Vision Spatial Reasoning.

Figure 2: **Different vision experts can provide *rich* visual prior knowledge**, which can be transferred to VL learning, and efficiently improve visual perception capability with limited, small-scale data.

wherein it incorrectly imagines details about an image. Given the availability of numerous pre-trained vision models from public repositories, our intuition is that **"why not fully utilizing these vision experts and transfer their knowledge to enhance the visual perception capability?"** As shown in Figure 2, different experts exhibit distinct vision properties for the same inputs, and each can contribute uniquely when their knowledge is transferred to VL learning (Geman et al., 1992).

To this end, we propose a VLM that **T**ransfers the knowledge from a hub **o**f **V**ision **E**xperts (**ToVE**) for efficient VL learning. Building on recent VLM designs (Liu et al., 2024b; Li et al., 2023a), where a frozen CLIP image encoder provides vision tokens for image-conditioned language generation, we establish a model hub that includes multiple domain-specific vision experts and a token-aware gating network that dynamically routes "expert knowledge" into every vision token. To preserve the generalizability of the vision tokens from CLIP, we develop a "*residual knowledge transfer*" strategy when transferring the knowledge to vision tokens. Since the experts are not coupled in the ToVE framework, we can selectively detach experts with minimal contributions to enhance inference efficiency based on their average gating weights during the training stage. Further, since the knowledge from vision experts acts as a complement or calibration for the vision tokens, we merge the expert knowledge to the vanilla CLIP vision encoder via the proposed "*knowledge merging*" approach. This approach eliminates the need for expert inference, significantly boosting inference efficiency while maintaining robust vision perception capabilities in VL tasks. In summary, our main contributions are as follows:

• **Token-aware Knowledge Transfer from Vision Experts.** Compared with previous works relying on large-scale models and datasets to develop the vision capabilities required by VL tasks from scratch, we construct a model hub from readily available vision experts, transferring their knowledge to VL tasks for efficient learning. The proposed ToVE can dynamically route the optimal vision knowledge to respective vision tokens and adopts a residual transfer strategy to enhance the original vision tokens while maintaining their generalizability. Consequently, ToVE can achieve competitive performance with two orders of magnitude less training data.

• **Pluggable Vision Experts and Knowledge Merging.** Since the vision experts in ToVE are not coupled and their knowledge serves to complement or calibrate each vision token, this allows us to selectively detach the low-contributing experts to improve inference efficiency. Furthermore, with the vision tokens enriched with expert knowledge, we introduce a "knowledge merging" approach to adapt this knowledge into a single vision encoder. This approach eliminates the need for vision expert inference while achieving promising performance without any vision experts.

## 2 RELATED WORK

**Vision-language Learning.** Vision-language (VL) learning represents the integration of visual and language processing capabilities. This field typically follows a dual-phase approach: pre-training

and task-specific fine-tuning. During the pre-training phase, models are trained on image-text pairs, enabling them to learn visual perception aligned with the texts, thereby enhancing performance in downstream VL tasks. Fine-tuning involves transferring this model knowledge to specific VL tasks, such as image captioning (Lin et al., 2014) and visual question answering (VQA) (Antol et al., 2015). Notably, the pre-training phase is data-intensive, often requiring billions of image-text pairs to achieve satisfactory performance (Wang et al., 2022a; Alayrac et al., 2022; Wang et al., 2023). Recent advancements have seen some studies propose efficient training methods that offer improved performance even with smaller models and fewer data requirements, such as Prismer (Liu et al., 2024b), MAMO (Zhao et al., 2023), and EVE (Chen et al., 2024a). The work closest to ours is Prismer (Liu et al., 2024b), which requires more data and additional training of a "Resampler" (Alayrac et al., 2022) to learn to implicitly synthesize expert knowledge into auxiliary vision tokens. In contrast, we aim to transfer the vision knowledge from the vision experts to the original vision tokens. The transfer phase is explicit and interpretable, with a token-aware gating network that dynamically routes expert knowledge to these vision tokens.

**Conditional Computation.** The process of knowledge transfer from vision experts is close to conditional computation (Yang et al., 2019; Chen et al., 2020; Han et al., 2021). Recent works introduce the idea into mixture-of-experts (MoE) (Riquelme et al., 2021; Fedus et al., 2022; Wang et al., 2024) within Transformer architectures (Vaswani et al., 2017), where multiple Feed Forward Networks (FFNs) serve as experts and a gating network selectively activates these experts to process the given input tokens. Different from these models which create mixture-of-experts from learnable FFNs with random initializations, we seek to transfer the knowledge from diverse, pre-trained vision experts to VL learning. Furthermore, MoE models typically prioritize load balancing (Fedus et al., 2022) to ensure the full utilization of each expert, while our approach focuses on adaptively learning the optimal assignment of experts to efficiently transfer the vision knowledge for various VL tasks.

**Learning from Models.** Given the abundance of pre-trained models (referred to as experts) trained on diverse datasets, learning from models aims to leverage the knowledge gained from existing models to enhance model performance, rather than training from scratch with raw data. Traditional methodologies, such as fine-tuning and knowledge distillation, are frequently used but often fail to fully utilize the knowledge from existing models. To address their limitations, various model merging techniques have been developed to amalgamate or edit the weights from different homogeneous models, such as model soup (Wortsman et al., 2022), task arithmetic (Ilharco et al., 2022), and DARE (Yu et al., 2023). Recent efforts have also explored the ensemble of multiple heterogeneous models to tackle tasks such as classification (Shu et al., 2022), domain adaptation (Li et al., 2022c), and language generation (Jiang et al., 2023). These approaches typically collect models from similar tasks and ensemble them at the task level. In contrast, we leverage vision experts from distinct domains, transferring their extensive and diverse vision knowledge to VL learning. In ToVE, we develop a "residual knowledge transfer" strategy to dynamically transfer optimal expert knowledge for each vision token, thereby enhancing vision token representations.

## 3 METHODOLOGY

### 3.1 THE OVERALL FRAMEWORK

The ToVE framework, as illustrated in Figure 3, integrates a CLIP (Radford et al., 2021) vision encoder $\mathbf{E}_{\text{vis}}$, a fine-grained gating network $\mathbf{G}$, a model hub with $K$ vision experts $\{\mathbf{E}_1, \mathbf{E}_2, ..., \mathbf{E}_K\}$ sourced from public repositories, and a language decoder $\mathbf{D}_{\text{lang}}$. For each image, its vision tokens are encoded by $\mathbf{E}_{\text{vis}}$, and are then sent to the gating network $\mathbf{G}$ to obtain the optimal assignment of expert knowledge at the token level. The expert knowledge is fused and transferred to the vision tokens via the proposed residual knowledge transfer strategy. After that, the knowledge-enhanced vision tokens are sent to the language model through cross-attention to condition the language generation.

### 3.2 TRANSFER KNOWLEDGE FROM A HUB OF VISION EXPERTS

**Expert Token Projector.** Since the vision experts vary in all the aspects including tasks, datasets, learning paradigms, and the network architectures, expert token projectors are required to *align these experts within a unified embedding space* before the knowledge transfer phase. The projector is a

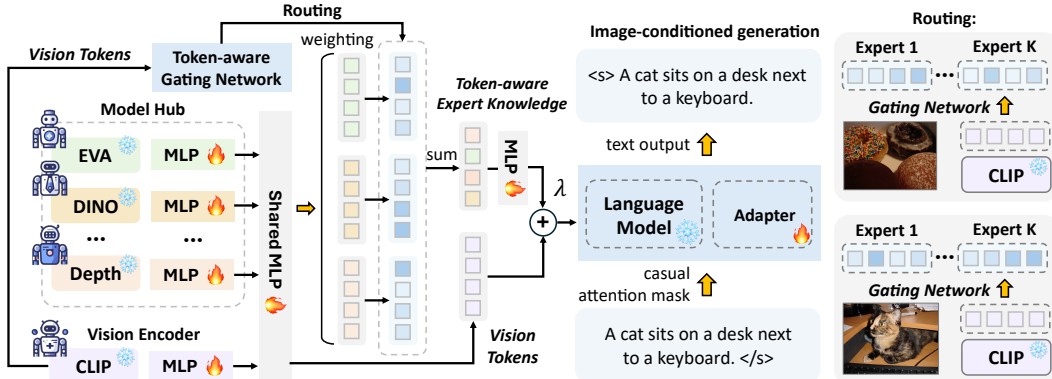

Figure 3: **The overall *framework* of ToVE.** The vision tokens processed by the vision encoder $\mathbf{E}_{\text{vis}}$ are assigned expert knowledge through the gating network, then enhanced with a "residual knowledge transfer" strategy before interacting with the language model. For the gating network, it dynamically assigns the optimal expert knowledge to each vision token for VL learning.

multi-layer perceptron network (MLP) with a GeLU (Hendrycks & Gimpel, 2016) non-linearity, where the first layer is expert-specific and the second layer is shared among all vision experts. Specifically, given the $k$-th ($k \leq K$) vision expert $\mathbf{E}_k$, its projector is parameterized as: $\psi_k = [\psi_1^k; \psi_2]$, where $\psi_1^k$ denotes the weights of the first layer specific to $\mathbf{E}_k$ and $\psi_2$ denotes the weights of the second layer shared across experts. The projection function of can be delineated as $\mathbf{H}_{\psi_k} \in \mathcal{A} : \mathbb{R}^{d_k} \to \mathbb{R}^{d_{\text{lang}}}$, where $d_k$ is the token dimension of the expert $\mathbf{E}_k$ and $d_{\text{lang}}$ is the token dimension of the language model $\mathbf{D}_{\text{lang}}$.

**Token-aware Expert Knowledge Ensemble.** The representation of each vision token is distinct, carrying distinct knowledge (e.g., foreground objects, depth, spatial positions) for complex VL tasks. This necessitates a specialized strategy to ***transfer unique expert knowledge to each token***. To achieve this, we introduce a token-aware gating network parameterized as $\theta$, with a routing function defined as $\mathbf{G}_\theta \in \mathcal{A} : \mathbb{R}^{d_{\text{vis}}} \to \mathbb{R}^K$, where $d_{\text{vis}}$ denotes the length of the vision tokens. For each vision token $\boldsymbol{t}_{\text{vis}}^i \in \mathbb{R}^{d_{\text{vis}}}$, the gating network takes it as input and computes its routing score $\boldsymbol{r}_i = [r_1, r_2, ..., r_k] \in \mathbb{R}^K$ for each expert. Different from previous MOE works (Riquelme et al., 2021; Fedus et al., 2022) that activates the expert with the top-1 routing score, we argue that knowledge from a single expert domain is insufficient, and, in fact, the gating network can adaptively learn the assignment of these experts. Therefore, we propose to apply ***an ensemble of the tokens $\boldsymbol{t}_k^i$ derived from the K vision experts*** to produce the expert knowledge for each vision token. Specifically, for the vision token $\boldsymbol{t}_{\text{vis}}^i$, we normalize its routing score $\boldsymbol{r}_i$ by softmax function $\text{Softmax}(\boldsymbol{r})_j = e^{r_j} / \sum_{k=1}^K e^{r_k}$, imposing a relative competition among the vision experts. That is, the final ensemble weight of expert knowledge is computed as $\boldsymbol{w}_i = \text{Softmax}(\boldsymbol{r}_i + \epsilon)$, where we empirically add a small noise $\epsilon$ sampled independently $\epsilon \sim \mathcal{N}(0, \frac{1}{K^2})$ entry-wise to improve the exploratory behavior of the gating network and the robustness of the model. Finally, the expert knowledge token for $\boldsymbol{t}_{\text{vis}}^i$ can be computed as:

$$\boldsymbol{t}_{\text{exp}}^i = \sum_{k=1}^K \left[ w_k \cdot \mathbf{H}_{\psi_k}(\boldsymbol{t}_k^i) \right], \text{ for } i = 1, 2, \ldots, N, \tag{1}$$

where $N$ is the total number of vision tokens.

**Residual Knowledge Transfer.** Transferring expert knowledge into vision tokens can be achieved through two straightforward strategies: (a) directly integrating $\boldsymbol{t}_{\text{exp}}$ into vision token via addition; (b) appending $\boldsymbol{t}_{\text{exp}}$ as the auxiliary vision tokens; The former strategy (a) preserves the count of vision tokens but may lead to an over-reliance on expert knowledge, potentially overwhelming the generalizable vision tokens from CLIP. In contrast, the latter strategy (b) introduces additional computational burden and increases the complexity of learning from these auxiliary expert tokens. To address the above limitations, we introduce a *residual knowledge transfer* method, inspired by the residual architectures employed in modern foundation models (He et al., 2016; Devlin et al., 2018).

For each vision token $\boldsymbol{t}_{\text{vis}}^i$, its expert knowledge $\boldsymbol{t}_{\text{exp}}^i$ is transferred via a *residual addition*, which is defined as follows:

$$\tilde{\boldsymbol{t}}_{\text{vis}}^i = \boldsymbol{t}_{\text{vis}}^i + \lambda \times \mathbf{M}_\phi(\boldsymbol{t}_{\text{exp}}^i), \tag{2}$$

where $\mathbf{M}_\phi(\cdot)$ denotes a two-layer MLP function : $\mathcal{A} : \mathbb{R}^{d_{\text{lang}}} \to \mathbb{R}^{d_{\text{lang}}}$, parameterized by $\phi = [\phi_1; \phi_2]$. Here, $\lambda$ is the coefficient to reconcile the proportion of expert knowledge transferred into original vision tokens. By incorporating expert knowledge as a residual addition term adjusted by $\lambda$ and an MLP function, rather than a direct alteration of the existing CLIP vision tokens, this strategy *seamlessly* integrates expert knowledge into the vision-language learning process, maintaining both the generalizability of the CLIP vision tokens and computational efficiency of the model.

**Pluggable Vision Experts.** From our analysis in Section 4.4, there is a positive correlation between the average ensemble scores $\boldsymbol{w}$ and the experts' contributions. Since ToVE does not couple expert knowledge in the transfer phase, it is easy to detach low-contributing experts from the architecture to improve the inference efficiency. Suppose we choose the top-$\tilde{k}$ contributing experts $\mathcal{E}$ ($|\mathcal{E}| < K$) for inference, the routing scores $\boldsymbol{r}_i$ of the detached vision experts are set to $-\infty$:

$$\boldsymbol{r}_i = f_{\text{topk}}^{\tilde{k}}\left(\mathbf{G}_\theta(\boldsymbol{t}_{\text{vis}}^i)\right), \text{where } f_{\text{topk}}^{\tilde{k}}\left(\mathbf{G}_\theta(\boldsymbol{t}_{\text{vis}}^i)\right)_k = \begin{cases} \mathbf{G}_\theta(\boldsymbol{t}_{\text{vis}}^i)_k & \text{if } \mathbf{E}_k \in \mathcal{E} \\ -\infty & \text{otherwise} \end{cases}. \tag{3}$$

After applying the softmax function, only the ensemble weight of top-$\tilde{k}$ contributing experts will be maintained and reconciled. Experts not included in $\mathcal{E}$ do not need to participate in the inference process. This operation allows the flexibility to detach any number of experts *without additional training*, achieving a balance between computational resources and model performance.

### 3.3 VISION-LANGUAGE LEARNING OF TOVE

**Language Modeling Pre-training.** Based on an image-text pair dataset $\{\text{I}, \text{T}\} \in \mathcal{D}$, ToVE is pre-trained with a unified language modeling loss without image-text contrastive and image-text matching losses commonly employed in other works (Li et al., 2021; 2022b). The loss of ToVE is formulated as follows:

$$\mathcal{L}_{\text{lm}}(\psi, \phi, \theta) = \mathbb{E}_{(\text{I},\text{T})\sim\mathcal{D}} \, \ell\left(\mathbf{D}_{\text{lang}}(\tilde{\boldsymbol{t}}_{\text{vis}}, \, p), \text{T}\right), \tag{4}$$

where $\tilde{\boldsymbol{t}}_{\text{vis}}$ is the knowledge-transferred vision tokens of image I, $\ell$ is the cross-entropy function between the predictions from the language decoder and the actual text description (i.e., caption) of the images, and $p$ is the prompt (i.e., prefix) of the language model. $\mathcal{L}_{\text{lm}}$ ensures the alignment between the knowledge-transferred vision tokens and text generation, leading the framework to explore optimal fusion configurations across diverse vision experts.

**Enhancing Exploration of Vision Experts.** When optimizing solely with $\mathcal{L}_{\text{lm}}$, the gating network $\mathbf{G}_\theta(\cdot)$ is prone to overfit and fall into a local minimum by *trivially assigning ensemble weights to a few specific vision experts*. One main reason is that different experts have distinct domain knowledge, and vary in the difficulty of transferring their knowledge to the vision tokens through $\mathbf{H}_\psi$. This issue makes ToVE difficult to explore other potentially valuable vision knowledge. Thus, to ensure the sufficient learning of $\mathbf{H}_\psi$ before exploring the optimal routing of the vision experts, we employ an auxiliary load balancing loss $\mathcal{L}_{\text{aux}}(\theta)$ on the gating network $\mathbf{G}_\theta$, followed by (Shazeer et al., 2017; Lepikhin et al., 2020; Fedus et al., 2022). The specifics of this implementation are detailed in Appendix A.3. During the training phase, the vision experts can gradually pro-

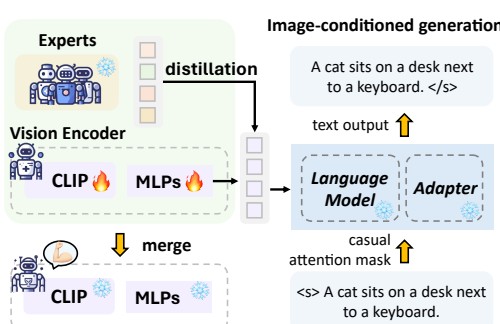

Figure 4: **The overview of expert knowledge merging.** The CLIP vision encoder enables the merging of expert knowledge into itself by predicting the knowledge-transferred vision tokens as an auxiliary learning target.

vide effective knowledge to vision tokens, hence, we set a relaxing coefficient $\alpha$ with a cosine schedule to progressively reduce $\mathcal{L}_{\text{aux}}$ to learn the optimal routing of vision experts: $\alpha_t =$

| Model | Pre-train (# pairs) | CIDEr | SPICE |
|---|---|---|---|
| SimVLM$_{\text{HUGE}}$ | 1.8B (600×) | 101.4 | - |
| BLIP[†] | 129M (43×) | 113.2 | 14.8 |
| BLIP-2$_{\text{OPT2.7b}}$[†] | 4M (1.3×) | 111.9 | 14.5 |
| Prismer$_{\text{LARGE}}$ | 12.7M (4.2×) | 107.9 | 14.8 |
| Prismer$_{\text{BASE}}$ | 12.7M (4.2×) | 87.5 | 13.0 |
| VinVL[†] | 5.7M (1.9×) | 95.5 | 13.5 |
| OSCAR[†] | 4M (1.9×) | 80.9 | 11.3 |
| ToVE (no experts) | 3M | 92.1 | 13.3 |
| ToVE-Lite | 3M | 104.1 | 14.4 |
| **ToVE** | **3M** | **110.2** | **14.9** |

Table 1: **Zero-shot caption performance on Novel Object Caption (NoCaps).** "†": the model is *fine-tuned* on COCO caption dataset, and then conduct zero-shot caption test on NoCaps.

| Models | VSR | POPE-R | POPE-A |
|---|---|---|---|
| MiniGPT-4 | 50.7 | 78.9 | 71.4 |
| LLaVA | 56.3 | 68.7 | 67.0 |
| BLIP-2$_{\text{V-7B}}$ | 50.0 | - | - |
| InstructBLIP$_{\text{V-7B}}$ | 54.3 | 89.3 | 78.5 |
| Prismer-Z (12.7M) | 63.2 | 84.9 | 81.2 |
| Prismer-Z[*] (3M) | 55.3 | 81.7 | 80.5 |
| ToVE-Lite (3M) | 65.9 | 86.6 | 81.9 |
| **ToVE (3M)** | **67.7** | **87.4** | **82.5** |

Table 2: **Zero-shot performance on VSR and POPE.** Both Prismer-Z and ToVE are fine-tuned on VQAv2, and tested in a zero-shot manner. *: the results are reproduced by us (using the same dateset as ToVE). V-7B: Vicuna-7B (Zheng et al., 2024); Accuracy is adopted on VSR, and F1-score is adopted on POPE.

$\alpha_0 \times 0.5 \times \left(1 + \cos\left(\pi \times \frac{t}{T}\right)\right)$, where $\alpha_t$ represents the coefficient at any given training epoch $t$, $T$ is the total training epochs, and $\alpha_0$ is the initial coefficient at the training start.

**Learning Objective.** With the incorporation of language modeling loss and the load balancing loss, the final optimization problem becomes:

$$\arg \min_{(\theta,\psi,\phi)} \mathcal{L}_{\text{lm}} + \alpha \cdot \mathcal{L}_{\text{aux}}. \tag{5}$$

Our ToVE framework simultaneously explores the knowledge transfer to vision tokens and vision-language learning, which can be trained in an end-to-end manner.

### 3.4 REDUCE ALL EXPERTS INTO ONE

As more experts are integrated into the model hub, the computation load is increasingly intensified. Therefore, inspired by distillation techniques that bridge the gap between training and inference (Gou et al., 2021), we propose to ***merge the expert knowledge*** to the CLIP vision encoder, dubbed "expert knowledge merging". The proposed "residual knowledge transfer" strategy is essential for effective knowledge merging, as it transfers the expert knowledge to the vision tokens without significantly altering the original representations. That is, there is a small representation gap $\boldsymbol{g} := \tilde{\boldsymbol{t}}_{\text{vis}} - \boldsymbol{t}_{\text{vis}}$ between the original vision tokens and the knowledge-transferred vision token, and the learning target of "expert knowledge merging" is to minimize this gap $\boldsymbol{g}$. As depicted in Figure 4, we maintain the language modeling loss while simultaneously merging the expert knowledge to the vision encoder through the l2-norm loss:

$$\mathcal{L}_{\text{transfer}} = \mathbb{E}_{(\text{I},\text{T})\sim\mathcal{D}} \left[ \ell\left(\text{D}_{\text{lang}}(\boldsymbol{t}_{\text{vis}}, p), \text{T}\right) + \|\underbrace{\tilde{\boldsymbol{t}}_{\text{vis}} - \boldsymbol{t}_{\text{vis}}}_{\boldsymbol{g}}\|_2 \right], \tag{6}$$

where $\tilde{\boldsymbol{t}}_{\text{vis}}$ and $\boldsymbol{t}_{\text{vis}}$ denote the original and the knowledge-transferred vision tokens, respectively. In this transfer stage, only the CLIP vision encoder with its MLPs are online updated, while other components in ToVE remain frozen. In inference stage, only the knowledge-merged CLIP provides the vision token for language generation.

## 4 EXPERIMENTS

### 4.1 IMPLEMENT DETAILS

ToVE utilizes ViT (Dosovitskiy et al., 2020) pre-trained by CLIP (Radford et al., 2021) as the frozen vision encoder, and RoBERTa (Liu et al., 2019) as the frozen language decoder, following the practice in (Liu et al., 2024b). Both of them have 12 (base-size) Transformer blocks. In the pre-training stage, the image resolution is set to $256 \times 256$. We use random resized cropping and horizontal

| Model | Pre-train (# pairs) | COCO | | NoCaps | |
|---|---|---|---|---|---|
| | | B@4 | S | Out | Overall |
| LEMON | 200M | 40.3 | 23.3 | 107.9 | 106.8 |
| BLIP | 129M | 39.4 | - | 105.7 | 110.0 |
| BLIP | 14M | 38.6 | - | 102.4 | 105.1 |
| Prismer-Z | 12.7M | 39.7 | 24.1 | 105.8 | 107.5 |
| Prismer | 12.7M | 40.1 | 24.1 | 111.7 | 109.1 |
| GIT | 10M | 40.4 | 23.0 | 89.6 | 96.6 |
| VinVL | 8.9M | 38.2 | 23.6 | 83.8 | 94.3 |
| OSCAR | 6.5M | 36.5 | 23.1 | 77.6 | 81.1 |
| ToVE-Lite | 3M | 39.5 | 24.1 | 108.2 | 106.7 |
| **ToVE** | **3M** | **40.3** | **24.5** | **113.1** | **112.5** |

| Model | Pre-train (# pairs) | test-dev | test-std |
|---|---|---|---|
| BEiT3 | 3.1B | 77.7 | - |
| BLIP | 129M | 78.2 | 78.2 |
| Prismer | 12.7M | 76.8 | 77.0 |
| OSCAR | 8.9M | 73.2 | 73.4 |
| MAMO | 4M | 76.1 | 76.2 |
| MaskVLM | 4M | 75.5 | - |
| ALBEF | 4M | 74.5 | 74.7 |
| Triple | 4M | 74.9 | 74.9 |
| ToVE-Lite | 3M | 74.1 | 74.0 |
| **ToVE** | **3M** | **75.4** | **75.8** |

Table 3: **Fine-tuned caption performance** on COCO (Karpathy split) and NoCaps (validation set). These are all base-size models. CIDEr is adopted on NoCaps. B@4: BLEU-4; S: SPICE.

Table 4: **Fine-tuned VQA performance** on VQA v2 (test set). These are all base-size VLM models. Accuracy is adopted to evaluate the VQA performance.

flipping for data augmentation. The pre-training dataset is composed of two in-domain datasets (i.e., COCO (Lin et al., 2014) and Visual Genome (Krishna et al., 2017)) and one web dataset (i.e., CC3M (Sharma et al., 2018)). The web dataset is filtered and re-captioned by a pre-trained image captioner (Li et al., 2022b). In the fine-tuning stage, the image resolution is increased to $480 \times 480$, and the load balancing loss is not employed to further exploit the optimal expert assignment to specific VL tasks. To enhance the model's capability to process knowledge-transferred vision tokens, lightweight MLP adaptors (Houlsby et al., 2019) are integrated within each transformer layer of the language model. More training details are provided in Appendix A.8.

## 4.2 VISION EXPERTS IN THE MODEL HUB

**Low-level Vision Experts.** ToVE is equipped with three low-level vision experts from the domains of depth (Ranftl et al., 2021), surface normal (Bae et al., 2021), and edge (Poma et al., 2020). These predicted labels are conducted patch embedding operations through randomly initialized convolutional layers. Specifically, we employ five convolutional layers with a small $[3 \times 3]$ kernel to encode their respective expert knowledge. The details of these vision experts and encoding convolution layers can be viewed in Appendix A.2.

**Embedding Vision Experts.** We include two embedding vision experts, i.e., DINO (Caron et al., 2021) and EVA (Fang et al., 2023), trained from self-supervised and image-text contrasting paradigms, respectively. Compared to the low-level vision experts, we utilize their patch tokens as the vision knowledge. To align the number of tokens with the vision encoder (i.e., CLIP) for knowledge transfer, we apply an interpolation operation to their patch tokens. The details of these vision experts can be viewed in Appendix A.2.

## 4.3 RESULTS ON VISION-LANGUAGE TASKS

**Zero-shot Performance on Novel Object Captioning.** By adopting a unified language modeling objective to pre-train ToVE, it can naturally generate descriptions for images without requiring fine-tuning, thereby enabling zero-shot generalization. In Table 1, we compare ToVE against several prior arts, including SimVLM (Wang et al., 2022b), BLIP/BLIP2 (Li et al., 2022b; 2023a), VinVL (Zhang et al., 2021), OSCAR (Li et al., 2020), and Prismer (Liu et al., 2024b). Despite using fewer data pairs and smaller models, ToVE consistently demonstrates superior zero-shot captioning abilities. For instance, under a model scale similar to Prismer_BASE, ToVE achieves a CIDEr score of **110.2** and a SPICE score of **14.9**, surpassing Prismer_BASE by a large margin (**+22.7** in CIDEr) while utilizing an approximately **4×** smaller pre-training dataset. For the variant of ToVE after expert knowledge merging, dubbed "ToVE-Lite", it outperforms the baseline model without any expert participation by a CIDEr score of **+12.0**, attaining **95.6%** of the original ToVE's performance. Additionally, the zero-shot performance of ToVE models often exceeds the *fine-tuned* performance of certain other VLMs, such as LEMON and BLIP. More results can be viewed in Appendix A.5.

**Fine-tuned Performance on COCO Caption, NoCaps, and VQAv2.** We summarize the fine-tuned captioning and VQA evaluations in Table 3 and Table 4, respectively. With a smaller training dataset, ToVE demonstrates strong results for both COCO and NoCaps benchmarks while using fewer image-text pairs. Especially for NoCaps, with out-domain samples with novel objects, we make significant improvements over the prior arts, with a CIDEr of **113.1**. In VQA, We additionally compare ToVE against several prior arts, BEiT3 (Wang et al., 2023), MAMO (Zhao et al., 2023), MaskVLM (Kwon et al., 2023), (Li et al., 2021), and Triple (Yang et al., 2022). ToVE also achieves comparable performance with the prior arts. This indicates that the knowledge transfer from various vision experts is primarily responsible for good robustness and generalization.

**Vision Perception Capabilities.** We evaluate the VLM's vision capabilities using the Visual Spatial Reasoning (VSR) (Liu et al., 2023) and POPE (Li et al., 2023b) benchmarks. VSR evaluates the reasoning about the relative positions of different objects, while POPE evaluates the "object hallucination" issue. From Table 2, when the dataset is scaled down from 12.7M to 3M, we can observe the recent efficient VLM, Prismer-Z (Liu et al., 2024b), shows a marked decline (**-7.9%** in VSR) in visual perception capabilities with smaller datasets, with a more severe issue of object hallucination. Conversely, when transferring the knowledge from vision experts to VL, ToVE significantly outperforms Prismer-Z with the same 3M dataset (**+12.4%** in VSR), with improvements in mitigating object hallucination (**+2.0%** in POPE-A). Additionally, our comparison with recent multi-modal large language models (including MiniGPT-4 (Zhu et al., 2023), LLaVA (Liu et al., 2024a), and InstructBLIP$_{\text{V-7B}}$ (Dai et al., 2024)) demonstrates that VLMs with large-scale pre-training generally has better visual perception capabilities. Notably, ToVE shows a substantial advantage over current MLLMs, especially in vision spatial reasoning.

## 4.4    ANALYSIS OF TOVE

**Visualization of the routing maps from the Gating Network G.** As presented in Figure 6, we visualize the routing maps of the images from the COCO caption dataset as determined by ToVE's gating network. It showcases the gating network's adeptness in learning instance-dependent fusion weights at the patch level, crucial for transferring the most beneficial vision knowledge from the vision experts to the vision tokens. For the low-level experts (columns 2-4), a consistent pattern is observable across their routing maps, where they predominantly enhance the areas around objects. This pattern indicates that these experts significantly enrich the CLIP tokens with *low-level information*, such as depth perception around objects, edges, and surface orientations, which bolsters the model's ability to perceive fine details. In contrast, for the embedding experts (columns 5-6), we notice distinct patterns in knowledge contributions: DINO primarily enhances *the patch tokens associated with the main objects in the captions*, whereas EVA contributes significantly to *the understanding of the image backgrounds*. More routing maps can be viewed in Appendix A.4.

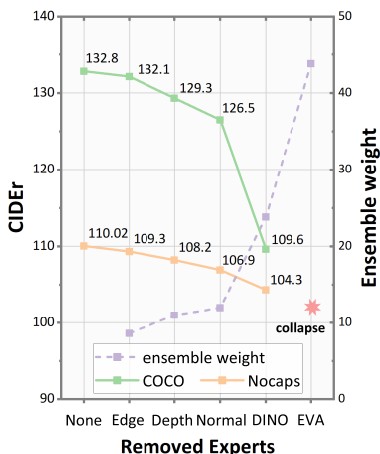

Figure 5: Impact of iteratively removing vision experts on **zero-shot caption performance**.

**Pluggable Vision Experts.** In Section 3.2, we explore the strategy of detaching less-contributing experts to enhance inference efficiency. It is built on a positive correlation between the ensemble scores $w$ and the performance benefits contributed by these vision experts. Based on the average ensemble scores, interpreted as "contribution", counted during the training phase, we *iteratively* remove experts in descending order of their average contributions. As depicted in Figure 5, it is observed that the performance declines noticeably as experts with increasingly significant contributions are removed. The model exhibits higher sensitivity to the in-domain dataset, COCO caption, compared to that on a hold-out dataset, NoCaps. Upon the complete removal of all experts, we notice a model collapse where the model fails in normal language generation. We attribute this collapse to the substantial discrepancy between the vision tokens without expert knowledge and the input space of the language model. We also conduct a similar experiment on the VQAv2 benchmark, the results can be viewed in Appendix A.6.

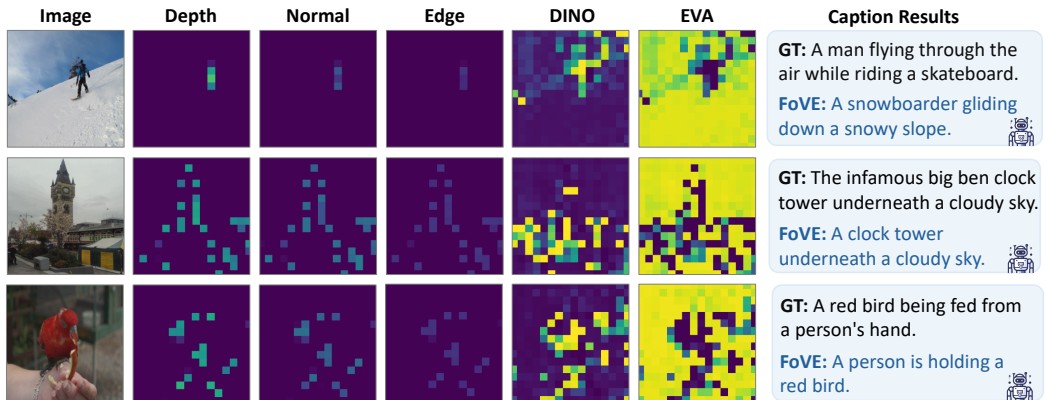

Figure 6: Visualization of **ToVE's routing maps** and the corresponding **caption results** on COCO caption without fine-tuning. Brighter patches indicate *higher* activation of the corresponding expert.

| Tasks | Baseline | Depth | Edge | Normal | DINO | EVA | ToVE |
|-------|----------|-------|------|--------|------|-----|------|
| COCO | 116.8 | 120.7 | 121.1 | 120.3 | 128.9 | 130.6 | **132.8** |
| NoCaps | 92.1 | 98.2 | 96.3 | 98.7 | 105.1 | 109.1 | **110.2** |
| VQAv2 | 70.0 | 70.2 | 70.1 | 70.1 | 72.9 | 74.4 | **75.8** |
| VSR | 54.8 | 59.8 | 60.1 | 61.0 | 65.3 | 63.8 | **67.7** |
| POPE-R | 80.7 | 85.9 | 86.4 | 86.3 | 85.8 | 85.7 | **87.4** |
| POPE-A | 80.3 | 80.7 | 80.8 | 81.2 | 82.0 | 80.8 | **82.5** |

Table 5: VL performance of the variants equipped with **different vision experts**. Baseline: the ablated variant without transferring any vision experts to VL learning.

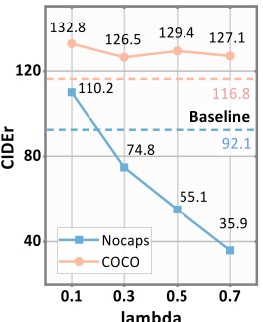

Figure 7: **Ablation study on** $\lambda$.

# 5 ABLATION STUDIES

**Different Vision Experts Transferred to VL Learning.** We evaluate the performance of each vision expert transferred to VL tasks. As shown in Table 5, we observe that each vision expert yields performance gains compared to the baseline without any vision experts. Specifically, the low-level experts significantly enhance visual perception capabilities, although their improvement in VQA, which requires strong multi-modal reasoning, is relatively marginal. On the other hand, the embedding experts contribute more substantially across all VL tasks, consistent with our average ensemble scores shown in Figure 5. DINO improves visual perception capabilities, demonstrating significant gains in VSR and POPE. Conversely, EVA promotes content understanding, showing notable performance improvements in captioning and VQA tasks. Also, we further compare the results of using EVA as the base vision encoder, the discussions can be viewed in Appendix A.10. After transferring the knowledge from all experts, ToVE can achieve significant performance enhancements across all VL tasks compared to the baseline without any experts, possessing both stronger visual perception and content understanding capabilities.

**$\lambda$ in Residual Knowledge Fusion.** $\lambda$ controls the proportion of transferring vision expert knowledge to the vision tokens. As depicted in Figure 7, our experiments demonstrate that a relatively small $\lambda$ results in superior performance compared to a larger one. For in-domain COCO captions, the proportion of expert knowledge fusion yields a relatively marginal impact. However, for NoCaps, which requires the model's generalizability on the novel objects, the model performance sharply declines (from **110.2** to **35.9**) as $\lambda$ increases beyond 0.1. This underscores the inherent strong generalization capabilities of CLIP for VL tasks. *An excessive fusion of expert knowledge, in turn, diminishes its generalizability.* Therefore, only a modest complement of expert knowledge is requisite for achieving satisfactory results.

**Knowledge Transfer and Merging.** In knowledge transfer strategies, we evaluate three approaches: (`A1`) direct addition of the expert knowledge tokens to the vision tokens, (`A2`) concatenation with the vision tokens, and (`A3`) residual addition to the vision tokens (ours). We observe that `A1` results in poor performance on NoCaps, which can be considered a special case where $\lambda$ is set to 1.0, as discussed in "$\lambda$ in residual knowledge fusion". While `A3` slightly underperforms in comparison to residual addition (`A3`), it does not deteriorate the generality of the CLIP model as `A1` does.

In knowledge merging strategies, we evaluate three approaches: (`B1`) direct distillation between the original vision tokens and the tokens with fused expert knowledge using L2-norm loss, (`B2`) language modeling to align the vision tokens with the input space of the language model that accepts the knowledge-transferred vision tokens, and (`B3`) a combination of language modeling and L2-norm distillation (ours). The results manifest that `B1` yields poor distillation performance, indicating a significant gap between the language model and the distilled CLIP. Although `B2` can di-

| Ablation | Strategies | NoCaps | COCO |
|---|---|---|---|
| Transfer | Direct addition | 26.8 | 126.3 |
| | Concatenation | 108.2 | 130.4 |
| | **Residual addition** | **110.2** | **132.8** |
| Merging | L2-norm | 10.5 | 45.6 |
| | LM | 96.2 | 118.7 |
| | **LM + L2-norm** | **106.7** | **128.6** |

Table 6: Ablation study on **knowledge transfer and merging strategies.**

rectly optimize the gap between the distilled CLIP and the language model, we can observe that combining language modeling with L2-norm (`B3`) achieves the optimal performance.

## 6 CONCLUSION

This paper proposes the ToVE framework for efficient vision-language learning by transferring knowledge from a hub of pre-trained vision experts. It utilizes a gating network with a residual knowledge transfer strategy to dynamically route expert knowledge to vision tokens, ensuring enhanced visual perception while preserving generalizability through residual knowledge transfer. This method allows for the selective detachment of low-contributing experts to improve inference efficiency. Additionally, we introduced a knowledge merging approach to merge expert knowledge into a single vision encoder. Experiments across various VL tasks demonstrate ToVE's competitive performance with significantly less training data, excelling in zero-shot captioning and visual spatial reasoning tasks. The visualization of gating network outputs and the analysis of pluggable vision experts highlight the effectiveness of transferring diverse vision knowledge to VL tasks, which presents a promising alternative to large-scale models and datasets.

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

# A APPENDIX

## A.1 MODEL ARCHITECTURE

Table 7 provides an overview of the architecture details for both ToVE-Lite and ToVE models. The vision encoder in both models utilizes a ViT-B/16 backbone with 12 layers. The language decoder is based on RoBERTa$_{BASE}$, also with 12 layers and a width of 768. For the ToVE-Lite model, the total number of trainable parameters is 91 million, with a total parameter count of 260 million when including non-trainable parameters. The ToVE model, which incorporates five vision experts, has 103 million trainable parameters. The total parameter count for the ToVE model, including the combined 1 billion parameters from the vision experts, sums up to 1.2B. These details illustrate the comprehensive design and the scalable nature of our models, accommodating varying levels of complexity and inference capabilities.

| Model Type | Vision Encoder | | Language Decoder | | | Trainable Params. | Total Params. |
|---|---|---|---|---|---|---|---|
| | Backbone | Layers | Backbone | Layers | Width | | |
| ToVE-Lite | ViT-B/16 | 12 | RoBERTa $_{BASE}$ | 12 | 768 | 91M | 260M |
| ToVE | ViT-B/16 | 12 | RoBERTa $_{BASE}$ | 12 | 768 | 103M | 1.2B |

Table 7: **ToVE-Lite and ToVE architecture details.** We detail the backbone, number of layers, and width for each architecture size, as well as the trainable and total parameters. For data inference, we include the total parameters, which encompass five vision experts with a combined parameter size of 1B in our ToVE model.

## A.2 VISION EXPERTS

The details of the vision experts are provided in Table 8. For ***low-level vision experts***, the expert labels are initially processed using randomly initialized convolutional layers to encode their respective vision knowledge. Specifically, we employ five convolutional layers with a small $[3 \times 3]$ kernel, which demonstrates superior performance compared to a single layer with a larger kernel, as evidenced in the Vision Transformer (Dosovitskiy et al., 2020). For ***embedding experts***, we select their large-size models. They have a patch size of 14, with input images sized at 224, producing 256 patch tokens. To achieve the residual knowledge transfer, their token quantity matches that of the vision encoder, which uses the Base-size model with a patch size of 16 and input images sized at 256.

| Task | Dataset | Model | Params. | Post-Processing |
|---|---|---|---|---|
| Depth Estimation | MIX-6 | DPT (Ranftl et al., 2021) | 123M | Re-normalised to $[-1, 1]$ and use convolution layers to encode the vision knowledge |
| Surface Normal | ScanNet | NLL-AngMF (Bae et al., 2021) | 72M | |
| Edge Detection | BIPED | DexiNed (Poma et al., 2020) | 35M | |
| Self-supervised | LVD-142M | DINO-v2 (Caron et al., 2021) | 304M | The input image size is resized to $224 \times 224$ and encoded to 256 tokens. |
| MIM + CLIP | Merged-38M | EVA-CLIP 02 (Fang et al., 2023) | 430M | |

Table 8: **Selected Tasks and Vision Experts** with Parameters and Post-Processing Techniques.

## A.3 LOAD BALANCING LOSS

To encourage a balanced assignment of vision tokens across different experts, we incorporate an auxiliary loss into the gating network. ***This auxiliary loss is beneficial to ensure the sufficient learning of $\mathbf{H}_\psi$ before exploring the optimal routing of the vision experts.*** Its consists of two parts: Importance loss and Load loss. The importance of the $k$-th expert is defined as the normalized

gating routing scores corresponding to the $k$-th expert, summed over the vision tokens $\boldsymbol{t}_{\text{vis}}$ of an image I:

$$\text{Imp}_k(\boldsymbol{t}_{\text{vis}}) = \sum_{\boldsymbol{t}^i_{\text{vis}} \in \boldsymbol{t}_{\text{vis}}} \texttt{Softmax}((\boldsymbol{G}_{\boldsymbol{\theta}}(\boldsymbol{t}^i_{\text{vis}}))_k. \tag{7}$$

The importance loss over the vision tokens $\boldsymbol{t}_{\text{vis}}$ from an input image I is defined as:

$$\mathcal{L}_{\text{imp}}(\boldsymbol{t}_{\text{vis}}) = \left( \frac{\text{std}(\text{Imp}(\boldsymbol{t}_{\text{vis}}))}{\text{mean}(\text{Imp}(\boldsymbol{t}_{\text{vis}}))} \right)^2. \tag{8}$$

In addition to the importance loss, we also introduce a load loss to ensure balanced routing results. The load of an expert $k$ given a vision token $\boldsymbol{t}^i_{\text{vis}} \in \boldsymbol{t}_{\text{vis}}$ is defined as the ***probability (frequency)*** of routing to expert $k$, summed over the vision tokens $\boldsymbol{t}_{\text{vis}}$ of image I:

$$\text{Load}_k(\boldsymbol{t}_{\text{vis}}) = \sum_{\boldsymbol{t}^i_{\text{vis}} \in \boldsymbol{t}_{\text{vis}}} p_k(\boldsymbol{t}^i_{\text{vis}}), \tag{9}$$

$$p_k(\boldsymbol{t}^i_{\text{vis}}) \triangleq P((\boldsymbol{G}_{\boldsymbol{\theta}}(\boldsymbol{t}^i_{\text{vis}}))_k \geq threshold_k(\boldsymbol{G}_{\boldsymbol{\theta}}(\boldsymbol{t}^i_{\text{vis}})). \tag{10}$$

The load loss over the vision tokens $\boldsymbol{t}_{\text{vis}}$ from an input image I is defined as:

$$\mathcal{L}_{\text{load}}(\boldsymbol{t}_{\text{vis}}) = \left( \frac{\text{std}(\text{Load}(\boldsymbol{t}_{\text{vis}}))}{\text{mean}(\text{Load}(\boldsymbol{t}_{\text{vis}}))} \right)^2. \tag{11}$$

The total auxiliary loss of the gating network is then given by:

$$\mathcal{L}_{\text{aux}} = \frac{1}{2} \left( \mathcal{L}_{\text{imp}} + \mathcal{L}_{\text{load}} \right). \tag{12}$$

### A.4 ROUTING OF THE GATING NETWORK

Figure 6 provides additional visualizations of the routing maps generated by ToVE's gating network for images from the COCO caption dataset. These maps highlight how the gating network assigns expert knowledge at the patch level. Low-level experts (Depth, Normal, Edge) predominantly enhance object-related areas, enriching the CLIP tokens with essential visual information. Embedding experts (DINO, EVA) show distinct contributions, with DINO focusing on main objects and EVA enhancing background understanding. These visualizations demonstrate the gating network's adaptive capability in optimizing vision token enhancement.

### A.5 VISUALIZATION OF CAPTION AND VQA RESULTS

Figure 10 illustrates the performance of ToVE and ToVE-Lite on zero-shot image captioning tasks using the NoCaps dataset and fine-tuned VQA tasks on VQAv2. Both models perform well, but ToVE consistently provides more detailed and contextually rich descriptions. For instance, ToVE specifies "hot chocolate" instead of just "coffee" and adds context like "being loaded onto a flatbed trailer." In VQA tasks, ToVE consistently delivers accurate and concise answers, demonstrating superior visual and language understanding capabilities. However, it is important to note that ToVE requires higher inference costs due to the integration of multiple vision experts. In contrast, ToVE-Lite, while slightly less detailed, still performs admirably with lower computational overhead. This trade-off between performance and inference cost should be considered when choosing between the two models based on the specific requirements of the application.

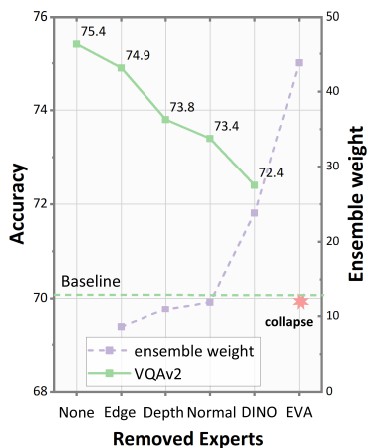

Figure 8: Impact of sequentially removing vision experts on **VQA**.

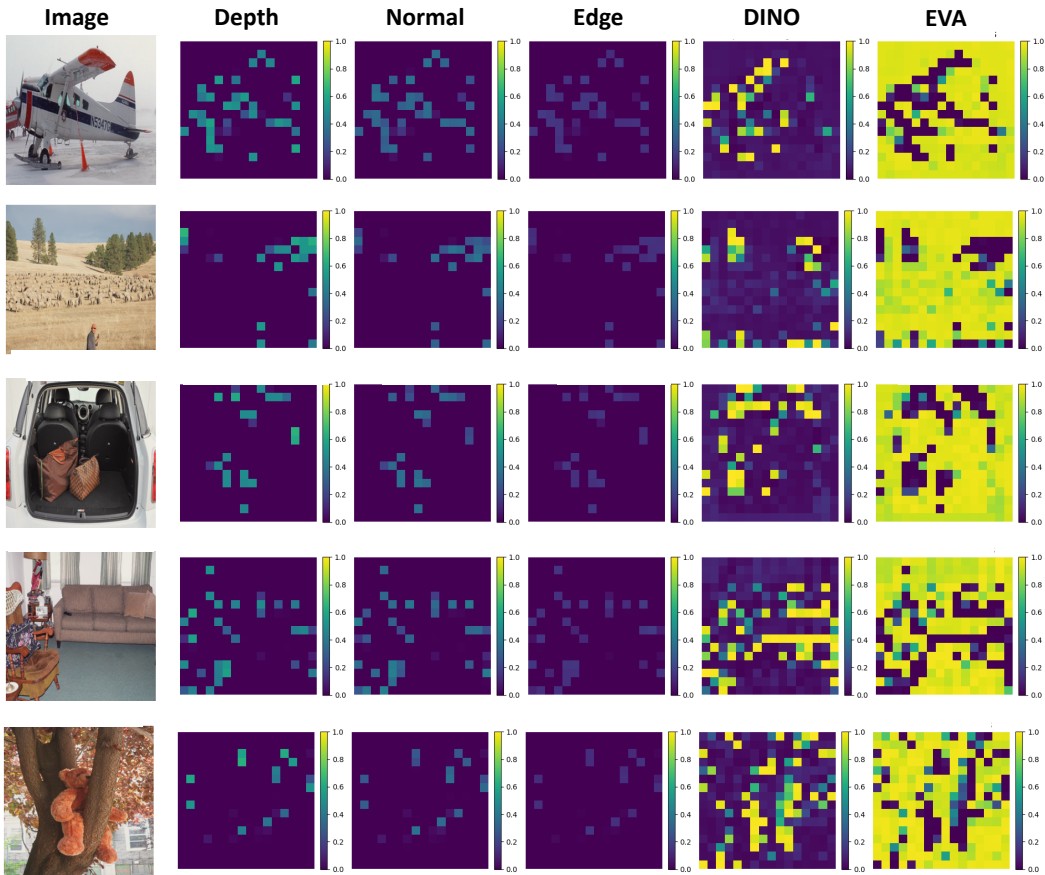

Figure 9: Visualization of **ToVE's routing maps** and the corresponding **caption results** on COCO caption without fine-tuning. Brighter patches indicate *higher* activation of the corresponding expert.

## A.6 PLUGGABLE VISION EXPERTS ON VQA

In further analysis, we investigate the impact of sequentially detaching experts on the visual question answering task using the VQAv2 dataset. The green line in Figure 8 illustrates the performance trajectory as vision experts are iteratively removed in descending order of their average ensemble weight, interpreted as their relative contribution. This trend mirrors the observations made on the image captioning task, underscoring the positive correlation between an expert's ensemble weight and its performance utility across diverse vision-language tasks. Notably, the degradation appears even more severe for VQAv2 compared to the COCO caption benchmark, implying a heightened reliance on the visual experts for this question answering challenge.

## A.7 LOAD BALANCING LOSS AND NOISE $\epsilon$

In our exploration of strategies to enhance the effective use of vision experts, we evaluate the impact of load balancing loss and the introduction of noise $\epsilon$ in Table 9. The results show that both load balancing loss alone (C1) and noise $\epsilon$ alone (C2) improve performance. Load balancing loss enhances performance on NoCaps to 109.9 and COCO to 132.1 by promoting balanced expert utilization. Noise $\epsilon$ improves performance to 109.4 on NoCaps and 130.9 on COCO by preventing overfitting to specific ex-

| Ablations | NoCaps | COCO |
|---|---|---|
| Load Balancing | 109.9 | 132.1 |
| Noise | 109.4 | 130.9 |
| **Load Balancing + Noise** | **110.2** | **132.8** |

Table 9: Ablation study on **Load Balancing Loss and noise $\epsilon$**.

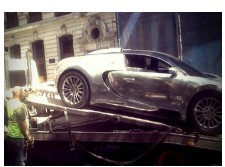

**ToVE-Lite:**
The car is on a flatbed trailer.
**ToVE:**
A silver car **being loaded onto** a flatbed trailer.

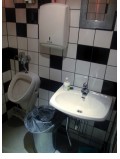

**Q:** What does the dispenser on the wall give? **ToVE: Toilet paper.**

**Q:** What colors are the tile on the wall? **ToVE: Black and white.**

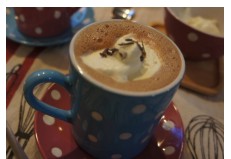

**ToVE-Lite:** A cup of coffee with whipped cream on top.

**ToVE:** A cup of **hot chocolate** with whipped cream on top.

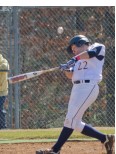

**Q:** What is the players number? **ToVE: 22.**

**Q:** Did he hit the ball? **ToVE: Yes.**

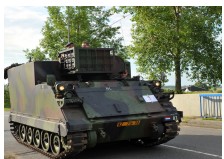

**ToVE-Lite:** A military tank driving down the road.

**ToVE:** A military tank driving down **a street next to trees**.

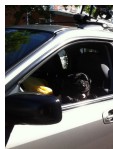

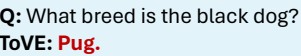

**Q:** Can the dog drive? **ToVE: No.**

**Q:** What breed is the black dog? **ToVE: Pug.**

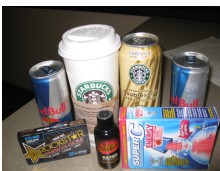

**ToVE-Lite:** A table with **a bunch** of drinks and a box of coke.

**ToVE:** A table topped with a cup of coffee and **a pack of starbucks energy drinks**.

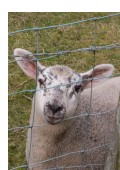

**Q:** Where is this animal commonly found?
**ToVE: Farm.**
**Q:** What color is it's nose?
**ToVE: Black.**

Figure 10: **Visualisation of zero-shot image captioning on NoCaps and fine-tuned VQA results on VQAv2.** ToVE can produce more detailed and semantically coherent captions than ToVE-Lite. We show a failure case (yellow box) of Nocaps dataset.

perts. Combining both strategies (C3) yields the best results, with 110.2 on NoCaps and 132.8 on COCO. Without load balancing, the gating network quickly converges to a trivial solution, predominantly routing to the EVA expert, making results similar to using EVA alone. Thus, the combination of load balancing and noise ensures a more effective and comprehensive integration of diverse expert knowledge, leading to more robust and generalized vision-language representations.

## A.8 TRAINING DETAILS

All our models are trained using the AdamW optimizer with a weight decay of 0.05. Automated data augmentation (AutoAug) is applied during both the pre-training and fine-tuning stages. For pre-training, the learning rate is set to 3e-4 with a total of 10 epochs. During fine-tuning for VQA, we use a learning rate of 1e-5 and train for 10 epochs. For fine-tuning the captioning model, the learning rate is 1e-5 with a total of 3 epochs.

## A.9 TOVE WITH LLMS

With the rapid advancements in LLMs, numerous LLM-based Vision-Language Models (LVLMs) have emerged. In this work, we also extend ToVE to language models to explore its potential in the LVLM domain. Specifically, we implemented ToVE within the LLaVA-1.5 framework (Liu et al., 2024a) (dubbed as "**ToVE_Vicuna**") with LLaVA's training data. During our implementation, we randomly sampled three-quarters of the dataset and utilized the entire instruction-tuning dataset. In the ToVE design, we integrated two domain experts—DINO (Caron et al., 2021) and Depth (Ranftl et al., 2021)—and employed QLoRA (Dettmers et al., 2024) to reduce computational overhead. To evaluate ToVE_Vicuna, we conducted experiments on the MME (perception) (Fu et al., 2023) and MMStar (perception) (Chen et al., 2024b) benchmarks to validate the improvement in visual capabilities brought by expert knowledge. As shown in Table 10, the results demonstrate a significant enhancement in perception capabilities through knowledge transfer. For instance, the MME perception score increased from 1434.5 to 1523.1, while the fine-grained perception subset of MMStar saw

| Models (QLoRA) | MME_p | MMStar (Overall) | MMStar (Coarse) | MMStar (Fine-grained) |
|---|---|---|---|---|
| LLaVA-1.5-7B | 1434.5 | 34.6 | 61.6 | 27.6 |
| ToVE_Vicuna | **1523.1** | **35.8** | **64.0** | **31.2** |

Table 10: Performance comparison of ToVE_Vicuna and LLaVA baseline on MME and MMStar.

an improvement of 3.6 points. These findings support the potential of ToVE in the LVLM domain. In the future, we will conduct further exploration in this direction.

| Benchmark | CLIP as Encoder | EVA as Encoder | CLIP + EVA expert |
|---|---|---|---|
| NoCaps | 92.1 | 95.7 | 109.1 |
| VQAv2 | 70.0 | 70.5 | 74.4 |
| VSR | 54.8 | 51.7 | 63.8 |

Table 11: Comparison of performance using different encoders.

## A.10 COMPARISON OF BASE VISION ENCODERS

We compared the performance of ToVE when using CLIP and EVA as the base vision encoders, with results summarized in Table 11. As shown, the two models exhibit comparable performance. Interestingly, we observed a notable improvement when combining CLIP as the vision encoder and EVA as the expert. We reckon that this enhancement can be attributed to EVA's capability to effectively process background representations, as evidenced by Figures 6 and 9. The visualizations illustrate the gating maps for each vision expert, revealing that EVA's gating activations predominantly occur in background regions, while showing minimal activation in subject areas. This observation suggests that, while low-level experts and DINO primarily focus on visual perceptual knowledge (as supported by their performance in visual perception tasks), their contributions to understanding background context remain limited. In contrast, EVA significantly improves semantic comprehension of these regions for the base vision encoder, thereby enhancing overall model performance.

## A.11 LIMITATIONS

Despite its advancements, the ToVE framework has several limitations. It heavily depends on the availability and quality of pre-trained vision experts, which may not always be available for certain domains or tasks. The initial integration and training with multiple experts can be computationally intensive, posing challenges in resource-constrained environments. Estimating the contribution of each expert relies on empirical methods that might not always yield optimal configurations, potentially affecting performance. While ToVE demonstrates robust generalization within its pre-training datasets, its effectiveness on completely unseen domains may vary, necessitating additional fine-tuning. Furthermore, the complexity of integrating multiple experts and dynamic mechanisms adds to the implementation and debugging challenges, especially for practitioners with limited experience in multi-modal learning frameworks. Addressing these limitations in future work could involve developing more adaptive methods for expert integration, optimizing computational efficiency, and enhancing generalization across diverse tasks and domains.

