# OpenReview forum: "ToVE: Efficient Vision-Language Learning via Knowledge Transfer from Vision Experts"
_ICLR.cc/2025/Conference — ICLR 2025 Poster_

### Official Review · Reviewer_6BHS · 2024-10-20

**Soundness:** 2
**Presentation:** 2
**Contribution:** 2
**Rating:** 6
**Confidence:** 3

**Summary:**

This paper proposes a method called ToVE, which fuses visual information from different visual encoders as the input of language model in a vision-language model. Specifically, the authors use CLIP-ViT as the main input and use it to weight the tokens from other backbones. Furthermore, the authors proposed a distillation algorithm to teach knowledge from different visual backbones to CLIP-ViT. Experiments show that the proposed method can bring certain improvements to the performance.

**Strengths:**

1) The proposed method is easy to understand.

**Weaknesses:**

1) The organization of this paper is somewhat confusing, especially the methods section.
2) The computation overhead can be siginifcantly higher than other methods.
3) The improvement of model capabilities may come from the introduction of a stronger visual backbone, rather than each model playing its own role in its professional field.

**Questions:**

1) This article is confusing in several ways:

    a) The author needs to further explain why setting the weights of tokens from unnecessary backbones to -inf in Formula 3 can improve computational efficiency. And authors should clarify if/how they are able to avoid activating all experts.

    b) In the section ``Enhancing Exploration of Vision Experts,'' the authors use the concept of L_aux but only cite it, lacking an explanation of how to apply it in this method. The authors should provide some explanation in the main text.

2. The motivation behind the manuscript and the source of the proposed method's performance need further clarification. From Fig. 8 and Fig. 9, it appears that the strong visual backbone, EVA, plays the primary role in most scenarios. Have the authors considered re-running the baseline experiment using EVA as the sole visual feature extractor? This will help clarify the contribution of the other experts beyond EVA and provide deeper insights into the model's overall performance.

3. There are some methods that ToVE should compare with:

    [1] Zi-Yi Dou, Yichong Xu, Zhe Gan, et.al.. An Empirical Study of Training End-to-End Vision-and-Language Transformers. CVPR 2022.

    [2] Wonjae Kim, Bokyung Son, Ildoo Kim. ViLT: Vision-and-Language Transformer Without Convolution or Region Supervision. ICML 2021

---

> ### Author Response · Authors · 2024-11-18
>
> We appreciate the reviewer's comments. Here, we resolve each of your concerns below.
>
> ---
>
> > The author needs to further explain why setting the weights of tokens from unnecessary backbones to -inf in Formula 3 can improve computational efficiency. And authors should clarify if/how they are able to avoid activating all experts.
>
> Thank you for your feedback. We would like to **it is NOT the case that setting the gating value of a specific expert to -inf directly improves computational efficiency**. INSTEAD, the computational efficiency is **achieved by detaching the low-contributing expert entirely from the architecture during inference**, as these experts are not deeply coupled in ToVE. This is the reason for the improved computational efficiency, as detailed in **Lines 226–229**.
>
> Regarding the rationale for setting the gating value of detached experts to -inf, this strategy **ensures proper reconciliation of ensemble weights** during the SoftMax operation.  When detaching Expert N, the gating network will still assign a gating score to Expert N. Assigning a value of -inf is to ensure that **its ensemble weight becomes 0** during the expert knowledge fusion process (**please see Lines 189–201 for details**).
>
> ---
>
> > In the section ``Enhancing Exploration of Vision Experts,'' the authors use the concept of L_aux but only cite it. The authors should provide some explanation in the main text.
>
> Thank you for your feedback. We would like to clarify that the details of L_aux within ToVE **have been elaborated in Appendix A.3**. The decision to exclude it from the main text stems from its role as an auxiliary learning loss in ToVE, which is **not the primary focus of our contributions**. We hope for your understanding on this matter since these has limited space in the main paper.
>
> ---
>
> > Have the authors considered re-running the baseline experiment using EVA as the sole visual feature extractor? This will help clarify the contribution of the other experts beyond EVA and provide deeper insights into the model's overall performance.
>
> Thank you for this insightful comment. In our early experiments, we have tested using EVA as the base vision encoder of ToVE. The results are presented below. As observed, **the performance of CLIP and EVA as standalone vision encoders is comparable**. Moreover, there are some works that adopt CLIP as the vision encoder, such as Prismer and BLIP. For the above reasons, we selected CLIP as the base vision encoder for ToVE.
>
> | dataset | CLIP as Encoder | EVA as Encoder | CLIP + EVA expert |
> | ------- | --------------- | -------------- | ----------------- |
> | NoCaps  | 92.1            | 95.7           | 109.1             |
> | VQAv2   | 70.0            | 70.5           | 74.4              |
> | VSR     | 54.8            | 51.7           | 63.8              |
>
> We also explored the reasoning behind the substantial performance improvement observed when combining CLIP and EVA as you noted in our experiments. In Table 5, **the contribution of EVA expert (CLIP + EVA) is mainly in the tasks that require semantic understanding**, such as caption and VQA. We attribute this improvement to EVA’s ability to effectively process background representations.
>
> Figures 5 and 9 provide visualizations of the gating map for each vision expert, where **EVA’s gating activations are predominantly observed in background regions**, with minimal activation in subject areas. This finding suggests that while low-level experts and DINO focus on visual perceptual knowledge (this is also supported by the results in the visual perception tasks), their contributions to background context understanding are limited. In contrast, EVA **enhances semantic comprehension in these regions for the base vision encoder**, benefiting the overall performance.
>
> ---
>
> > There are some methods that ToVE should compare.
>
> Thank you for highlighting these relevant works. We acknowledge their significance and appreciate your suggestion. In the revised version of the paper, we will include a comparison with the mentioned methods ([1] and [2]) to further enhance the completeness and quality of our paper.

---

> ### Author Response · Authors · 2024-11-21
>
> Dear reviewer, we want to kindly follow up on the submitted rebuttal. Given the importance of your feedback in refining and improving the work, we would greatly appreciate it if you could review the rebuttal at your earliest convenience.

---

> > ### Comment · Reviewer_6BHS · 2024-11-22
> >
> > Thanks for your reply, you solved my doubt and I am willing to improve my score.

---

> > ### Comment · Reviewer_6BHS · 2024-11-22
> >
> > I paid attention to the experiment results on LLaVA commented by Reviewer q7GH, and I think this result really proves the effectiveness of the method, so I am willing to further improve my score.

---

### Official Review · Reviewer_B56D · 2024-11-03

**Soundness:** 3
**Presentation:** 2
**Contribution:** 3
**Rating:** 6
**Confidence:** 5

**Summary:**

This paper presents a novel way to assimilate knowledge from different visual experts trained for different visual tasks into a pre-trained ViT  for solving visual language tasks. The idea of expert knowledge assimilation might not be new, but it is not trivial to align depth, surface normal or edge information along with a clip-based ViT model for vision language tasks. The authors achieve it by a carefully constructed architecture pipeline, where the experts and the ViT model are frozen, a token-based gating network selects which expert to distil knowledge from into that visual token, and then a language decoder utilizes these modified tokens for the final task. They further go on to merge all the expert knowledge into their vision encoder to relieve using multiple experts during inference.

**Strengths:**

1. The idea of fusing knowledge from multiple experts into the vision encoder is good.

2. The architecture design is novel, relating knowledge fusion to mixture of experts idea.

3. Although similar to Prismer in idea, ToVE does have better results

4. Ablations about types of knowledge merging, expert detachment are quite interesting and insightful

**Weaknesses:**

1. There are quite a few grammatical and sentence-construction mistakes throughout the paper, for e.g. "The projection function of can be delineated as" this sentence doesn't make sense. Also, the mapping function from d_k to d_lang should be about the feature dimension, since this is achieved by projection function MLPs, but it is written as token lengths, which is not the same as token feature dimension. Is this a writing mistake or the authors are changing token lengths, i.e., number of tokens?

2.  "Different from MOEs, which commonly activates the expert with the top-1 routing score," this is not true. The "original" MoE paper (Shazeer et al) did not have top -1, mainly switch transformer has top-1, and sparse MoE has top-k. Also, the authors employ a load balancing loss exactly similar to the Shazeer et al paper. So, stating that is a bit misleading.

3. After eq 6, it says t_clip and t_fuse, which does not even appear in the equation.

4. The method requires token-specific output from each expert (eq 1). So, how are tokens obtained for an image from low-level vision experts?

5. In Table 1 and 2, ToVE is not sota but its results are marked bold. (e.g. BLIPv2 CIDEr score in Table1, InstructBLIP POPE-R score in table 2).  Since comparison is tough due to architecture modification, expert merging, it is essential to atleast show number of training parameters to understand the comparison validity

**Questions:**

1. The paper is not written that carefully. I would suggest the authors to properly rewrite the paper considering grammatical mistakes, notation inconsistencies, etc.

2. How is a token-level knowledge extracted from low-level vision experts? This is not described properly either in main paper or appendix.

3. What would be proper baselines to compare against, since BLIP and others do not use knowledge sharing from other experts? I think the tables need a bit of redesign.

Please address these comments as well as the weaknesses. I am inclined to accept this paper since I like the architecture but the paper needs quite a bit of rework.

---

> ### Author Response · Authors · 2024-11-18
>
> We appreciate the reviewer's feedbacks. Here, we resolve each of your concerns below.
>
> ---
>
> > There are quite a few grammatical and sentence-construction mistakes. The stating of top-1 routing is a bit misleading.
>
> We sincerely apologize for the grammatical errors, notation inconsistencies, and sentence construction issues in our paper, as pointed out in your review.
>
> Regarding the mapping function from $d_k$ to $d_{lang}$, we intended to **refer to the "token feature dimension"**, as you correctly pointed out in the review. We have realized that the use of "token lengths" was a confusing misrepresentation.
>
> For $t_{clip}$ and $t_{fuse}$, we are sorry for our mistakes. They should be ~$t_{vis}$ and $t_{vis}$, which **represents the the original vision tokens and the knowledge-transferred vision tokens**.
>
> We acknowledge that the statement "Different from MOEs, which commonly activates the expert with the top-1 routing score" was an overclaim. As you rightly pointed out, the original MoE paper (Shazeer et al.) did not use top-1 routing and sparse MoEs commonly use top-k. We appreciate your feedback on this and will revise the this statement.
>
> We will carefully revise the manuscript to address these kinds of issues comprehensively. Thank you for your valuable feedback and for bringing these matters to our attention.
>
> ---
>
> > How is a token-level knowedge extracted from low-level vision experts? This is not described properly either in main paper or appendix.
>
> Thanks for pointing this out. In We would clarify that we include the details of encoding the low-level information from these experts **in Line 320-323 (Main paper)** and **Line 730-734 (Appendix A2)**. Specifically, **the encoding process is  close to the patch embedding operation** in the standard ViT architecture. These low-level labels are processed **using randomly initialized convolutional layers to encode their respective vision knowledge**. Each expert is equipped with five lightweight convolutional layers with a small [3 × 3] kernel. In the revised version, we will incorporate more details into the main text to emphasize this point.
>
> ---
>
> > What would be proper baselines to compare against, since BLIP and others do not use knowledge sharing from other experts?
>
> Thank you for your thoughtful and constructive feedback. Our primary motivation in this paper is to **transfer visual knowledge of pre-trained vision experts to enable efficient vision-language learning**. The vision experts are a fundamental component of ToVE, where the objective is to utilize their expertise alongside small-scale datasets to achieve competitive performance. Therefore, we compared ToVE with models that do not incorporate knowledge sharing from external experts. We added the "training samples" as to emphasize the training (data) efficiency in Table 1-4.
>
> Regarding the concern about trainable parameters mentioned in your review, we would like to clarify that **ToVE is relatively parameter-efficient** compared to many works. We have supplemented the some analyses of "trainable parameters", as shown in the table below, which will be incorporated into the revised version of the paper.
>
> | Method    | Trainable Parameters | Training samples | Nocaps (CIDER) |
> | --------- | -------------------- | ---------------- | -------------- |
> | ToVE_lite | ~80M                 | 3M               | 108.2          |
> | ToVE      | ~100M                | 3M               | 112.5          |
> | GIT       | ~100M                | 10M              | 96.6           |
> | BLIP      | ~450M                | 14M              | 105.1          |
> | BLIP      | ~450M                | 129M             | 110.0          |
>
> Once again, thank you for your valuable feedback, which has greatly contributed to improving the clarity and comprehensiveness of our work.

---

> ### Author Response · Authors · 2024-11-21
>
> Dear reviewer, we want to kindly follow up on the submitted rebuttal. Given the importance of your feedback in refining and improving the work, we would greatly appreciate it if you could review the rebuttal at your earliest convenience.

---

> > ### Comment · Reviewer_B56D · 2024-11-25
> >
> > Thank you for addressing my concerns. I am not fully convinced with the comparisons, but I also understand that it is quite difficult to find exact baselines to compare with in this field at the moment. I like this paper and would like to see it get accepted.

---

### Official Review · Reviewer_q7GH · 2024-11-03

**Soundness:** 2
**Presentation:** 3
**Contribution:** 2
**Rating:** 6
**Confidence:** 4

**Summary:**

This paper introduces ToVE, a selection method for vision encoders in vision-language models. ToVE utilizes multiple vision encoders, each pre-trained differently, and uses a gating network to select their output tokens, which are then added to the tokens from a CLIP encoder as visual features. Notably, these vision encoders can be distilled into a single encoder to reduce inference computation.

**Strengths:**

1. This paper proposes a novel method for enhancing vision-language models by leveraging multiple vision encoders.
2. Experiments demonstrate the superiority of using multiple vision encoders.

**Weaknesses:**

1. My primary concern is the significance of this approach. Using multiple vision encoders will multiply the parameter count and computational cost, making comparisons with other base-sized VLMs unfair. On the other hand, if distillation is used to merge them into a single encoder, the resulting composite encoder does not performs obviously better than simply using EVA as the encoder.

| dataset  | EVA    | ToVE-lite |
| ----- | ----- | ----- |
| NoCaps | 109.1 | 104.1|
| VQAv2   | 74.4   | 74.0|
| VSR        |  63.8 | 65.9|
| POPE-R | 85.7 | 86.6|
| POPE-C | 80.8 | 81.9|
| average |82.76|82.50|


This suggests that using ToVE-lite might be less effective than carefully selecting a single, well-performing encoder.

2. As mentioned above, due to the difference in parameter counts, comparisons with other VLMs may be unfair. Additionally, although the amount of VL data used for pretraining is indicated, the vision encoders and language models themselves are trained on large datasets, enhancing their individual visual and linguistic capabilities, which in turn boosts multimodal performance. Therefore, the amount of pretraining data for both the vision and language models should also be specified.

3. The setting used in this paper seems somewhat outdated, as the current trend in VLMs is toward general-purpose multimodal LLMs, such as LLaVA. I recommend that the authors implement ToVE within the LLaVA setting to demonstrate its effectiveness in broader, more contemporary scenarios.

4. The datasets used for comparison are somewhat limited. I suggest adding more datasets, such as GQA, TextVQA, and VizWiz, to provide a more comprehensive evaluation. Additionally, Table 3 should also report the CIDEr score for COCO.

**Questions:**

See weaknesses above.

---

> ### Author Response · Authors · 2024-11-18
>
> We appreciate the reviewer's feedbacks. Here, we resolve each of your concerns below.
>
> ---
>
> > The significance of ToVE and performance comparision with other methods.
>
> Thank you for pointing out this concern. We would like to clarify that the motivation of this paper is to **transfer visual knowledge from pre-trained vision experts**, which have already acquired diverse visual understandings, **to vision-language learning**, thereby achiving efficient learning.
>
> Although it is true that incorporating vision experts in ToVE increases the parameter count, we emphasize that our training cost—measured in terms of training samples and computational resources—is **substantially lower than that of other comparable methods**.
>
> Additionally, **we proposed strategies such as ToVE_lite and detachment of experts to mitigate inference costs**. Notably, **ToVE_lite, which operates without any vision experts, achieves competitive performance** compared to other methods (a more detailed response regarding ToVE_lite is provided below). We hope this explanation addresses your concerns.
>
> ---
>
> > The effectivess of ToVE_lite
>
> We would like to clarify that the results referenced in the review (Table 5) **do not correspond to using a single visual encoder**. **Instead, they represent the performance achieved when EVA is utilized as a vision expert** (i.e., the base CLIP vision encoder is combined with the EVA expert). As shown in Table 1-2, ToVE-lite, which employs a single knowledge-transferred CLIP as the vision encoder, achieves competitive performance, effectively demonstrating its efficacy.
>
> To further address your concern, we have included additional results in the table below, where **only EVA is used as the vision encoder**. These results indicate that EVA does not exhibit significant advantages compared to using CLIP as the vision encoder. Moreover, **EVA performs worse than ToVE-lite**, which reinforces the effectiveness of our proposed approach.
>
> | dataset | CLIP as Encoder | EVA as Encoder | CLIP + EVA expert | ToVE-lite |
> | ------- | --------------- | -------------- | ----------------- | --------- |
> | NoCaps  | 92.1            | 95.7           | 109.1             | 104.1     |
> | VQAv2   | 70.0            | 70.5           | 74.4              | 74.0      |
> | VSR     | 54.8            | 51.7           | 63.8              | 65.9      |
>
> ---
>
> > The amount of pretraining data for both the vision and language models should also be specified.
>
> We apologize for any confusion caused by the definition of the pretraining cost. In our context, **pretraining data refers specifically to the datasets used exclusively during the development of ToVE**. This definition intentionally excludes the original datasets utilized in training the expert models, **adhering to common practices within the VLM community**. For example, recent VLMs (e.g., Prsimer, BLIP, and BLIP-2) typically leverage pre-trained base vision models (e.g., CLIP, SigLIP) and text models (e.g., BERT, Vicuna) **without considering these as part of the pretraining data**.
>
> Importantly, our objective is to **harness the knowledge encapsulated in expert models to minimize the data requirements for constructing VLMs**. We hope this clarification addresses the reviewer’s concern.
>
> ---
>
> > ToVE with LLM setting and a more comprehensive evaluation.
>
> Thank you for your valuable suggestion. In response, we have initiated experiments integrating ToVE into an LLM (the same setting as LLaVA). However, due to the substantial computational and time requirements associated with the training and fine-tuning stages, **these experiments are still ongoing**. We are committed to **providing updated results within the next few days**.
>
> We acknowledge your recommendation to include a broader range of QA datasets for evaluation. **We will attach the results after the ToVE x LLM is finished**. Furthermore, we will include the CIDEr score for COCO in Table 3 in the revised version as recommended. Thank you for your comments, which will help improve the completeness and relevance of our work.

---

> ### Author Response · Authors · 2024-11-21
>
> Dear reviewer, we want to kindly follow up on the submitted rebuttal. Given the importance of your feedback in refining and improving the work, we would greatly appreciate it if you could review the rebuttal at your earliest convenience.

---

> ### Author Response · Authors · 2024-11-21
>
> UPDATE:
>
> Dear reviewer, following your recommendation, we have conducted additional experiments incorporating ToVE with an LLM. Specifically, we implemented ToVE within the LLaVA-1.5 framework (ToVE x Vicuna) using LLaVA’s training data, where we randomly sampled 3/4 of the pretraining data and utilized the full instruction-tuning dataset. In the ToVE design, we introduced two experts (DINO and Depth) and employed QLoRA to reduce the computational burden.
>
> Due to the time constraints during the rebuttal phase, we were unable to perform extensive hyperparameter tuning and full experiments. As such, there still remains needs for further optimization, and we kindly ask for your understanding. Below are the preliminary experimental results. These demonstrate that **knowledge transfer significantly enhances the model’s perception capabilities compared to the original baseline**.
>
> | Models                   | MME_p  | TextVQA | MMStar (Overall) | MMStar (Coarse Perception) | MMStar (Fine-grained Perception) |
> | ------------------------ | ------ | ------- | ---------------- | -------------------------- | -------------------------------- |
> | LLaVA-1.5-7B (QLoRA)     | 1434.5 | 49.7    | 34.6             | 61.6                       | 27.6                             |
> | ToVE x LLaVA-1.5 (QLoRA) | **1523.1** | **50.4**    | **35.8**             | **64.0**                       | **31.2**                             |
>
> We acknowledge your observation regarding the current trend toward general-purpose multimodal LLMs and agree that such models hold significant value. However, we believe there remains practical merit in leveraging smaller language models. These models allow for efficient training tailored to specific tasks, such as image captioning and VQA, which are highly relevant for practical applications.
>
> Lastly, we sincerely thank you for your constructive comments. We recognize the importance of general-purpose multimodal LLMs and plan to explore them further in future work.

---

> ### Comment · Reviewer_q7GH · 2024-11-22
>
> Thank you for your response. I will raise the score, but I hope the author revises the manuscript to include these new results.

---

### Official Review · Reviewer_dYR6 · 2024-11-05

**Soundness:** 3
**Presentation:** 3
**Contribution:** 3
**Rating:** 6
**Confidence:** 4

**Summary:**

This paper presents the ToVE (Transfer of Vision Experts) framework, leveraging pre-trained vision models to enhance vision-language learning efficiency by transferring expert knowledge via a token-aware gating network, resulting in competitive performance on vision-language tasks with significantly reduced data requirements.

**Strengths:**

1. The ToVE framework introduces a novel approach for efficient vision-language learning by utilizing a hub of pre-trained vision experts. This method promotes the effective transfer of knowledge, addressing the challenge of limited data availability in specialized domains.

2. The experimental results demonstrate that ToVE achieves competitive performance across various vision-language tasks using significantly less training data compared to existing models.

3. The paper includes visualizations of the gating network's routing decisions, illustrating how expert knowledge is allocated across different image regions. This enhances interpretability, showing how ToVE leverages expert knowledge in a token-specific manner to improve performance on complex visual tasks.

**Weaknesses:**

1.  The process of routing expert knowledge to vision tokens, particularly the token-aware gating network, adds complexity. Although the authors propose methods for detaching low-contributing experts to improve efficiency, the initial setup and training remain computationally intensive, which may hinder practical application.

2.  The paper introduces a load-balancing loss to ensure a balanced use of experts, but the effectiveness of this loss in preventing over-reliance on certain experts is not extensively validated, leaving questions about how it affects the model’s performance and robustness.

3.  The paper emphasizes the efficiency benefits of transferring pre-trained expert knowledge, yet it lacks a detailed discussion on how ToVE handles potential domain mismatches between the knowledge of vision experts and specific downstream vision-language tasks, such as highly specialized applications.

**Questions:**

As shown in Weaknesses.

---

> ### Author Response · Authors · 2024-11-18
>
> We appreciate the reviewer's comments. Here, we resolve each of your concerns below.
>
> ---
>
> > **The efficiency of ToVE for initial setup and training.**
>
> We appreciate the reviewer’s comments regarding the computational complexity associated ToVE’s training process. We would like to clarify that our proposed method, ToVE, demonstrates significantly improved efficiency in training requirements compared to many widely used vision-language models, such as SimVLM, GIT, and BLIP. The table below provides **a comparative analysis of training data sizes and computational costs (PFLOPs Days)**:
>
> | Model  | Training Data | Training Cost (# PFLOPs Days) |
> | ------ | ------------- | ----------------------------- |
> | SimVLM | 1.8B          | 66.9                          |
> | GIT    | 0.8B          | 45.8                          |
> | BLIP   | 129M          | 22.2                          |
> | ToVE   | 3M            | 0.37                          |
>
> As shown, ToVE achieves substantial reductions in both training data size and computational cost. While we acknowledge that the process of routing expert knowledge to vision tokens (including experts, MLPs, gating networks) introduces some additional complexity, **it does not significantly diminish the computational efficiency brought by our vision knowledge transfer process**. We hope this clarification addresses the reviewer’s concerns regarding the practicality of our approach.
>
> ---
>
> > **The discussion about the contribution of the load balancing loss in the training stage.**
>
> Thank you for your valuable feedback. We apologize for the insufficient discussion of the load balancing loss. For ToVE, the implement of load balancing loss is essential as there is a high risk that **easily transferable experts dominate in the early stages of training in our early experiments**. This phenomenon is particularly obvious between low-level experts and embedding experts.
>
> For low-level experts, ToVE learns from scratch how to perform patch embedding on their outputs to convert low-information data into tokens for the knowledge transfer. Conversely, learning the MLP mapping for embedding experts is relatively easier. Due to **the varying difficulty of knowledge transfer across experts**, the gating network without load balancing loss **tends to converge prematurely on embedding experts during early training**. This leads to diminished gradient flow for low-level experts (due to gating operations), causing the model to underutilize their valuable low-level information finally.
>
> We list one group of early experiment results in the following table to clarify this point. For the case of adopting DINO + Depth experts, the gating score of Depth expert is close to 0 without the load-balancing loss, and **the final model performance degrades to that of using only the DINO expert**. Similar phenomena can be observed across other expert configurations.
>
> | Experts               | DINO  | DINO + Depth  | DINO + Depth + load balancing |
> | --------------------- | ----- | ------------- | ----------------------------- |
> | Average Routing score | -     | **0.99 vs. 0.01** | 0.74 vs. 0.26                     |
> | CIDER of COCO         | 128.9 | **128.6**         | 130.1                         |
>
> In the revised version of the paper, we will include these experimental results and a detailed discussion to highlight the critical role of load-balancing loss in improving model robustness and performance.
>
> ---
>
> > **The applicability in highly specialized domain.**
>
> Thank you for your insightful comment. While our current experiments may not comprehensively address the applicability of ToVE in highly specialized domains, **we have demonstrated that different pre-trained experts can effectively transfer valuable knowledge to specific vision-language tasks requiring diverse vision capabilities**. For instance, the DINO model is leveraged for spatial reasoning, while low-level experts contribute to object perception. These results are detailed in Table 5 of the paper.
>
> For future work, we plan to extend ToVE to more specialized domains, such as medical report generation. A potential application would involve the MIMIC-CXR dataset [1], which comprises approximately 300k chest radiographs. In this context, we intend to utilize segmentation and classification models, which are widely available, as domain-specific experts to evaluate the adaptability and effectiveness of ToVE in specialized scenarios.
>
> [1] MIMIC-CXR: A de-identified, publicly available database of chest radiographs with free-text reports.

---

> > ### Comment · Reviewer_dYR6 · 2024-11-25
> >
> > Thank you for your response. I will keep my original score.

---

> ### Author Response · Authors · 2024-11-21
>
> Dear reviewer, we want to kindly follow up on the submitted rebuttal. Given the importance of your feedback in refining and improving the work, we would greatly appreciate it if you could review the rebuttal at your earliest convenience.

---

### Author Response · Authors · 2024-11-23

UPDATES:
Thanks to all reviewers for their thorough review and valuable comments. We have uploaded a new revised version of ToVE which incorporates the discussions of the concerns by the reviewers. We deeply appreciate your time and effort in helping us improve our work.

---

### Meta-Review · Area_Chair_heGD · 2024-12-20

**Metareview:**

Summary: This paper proposed ToVE (Transfer of Vision Experts) that transfer multi-expert knowledge to a vision encoder via a token-aware gating network and residual mechanism, which significantly reduced training data requirements.

Main strengths: (1) The approach of  leveraging multiple vision encoders as experts and transfer the knowledge into a single vision encoder is novel. (2) Extensive experiments validated the effectiveness of ToVE. (3) The visualization of ToVE's routing maps is interesting and insightful to understand the machenism.

Major weaknesses: (1) Experiments incorporating ToVE with an LLM (e.g., LLaVA-style) was lacking in the original version, and were added during discussion. (2) The writing quality, including explanations, organization, and notations, has space to be improved. (3) The compared

This paper received four positive scores as final rating, i.e., 6, 6, 6, 6. The AC agreed with reviewers and recommend the paper as accept.

**Additional Comments On Reviewer Discussion:**

The main concern addressed during discussion is incorporating ToVE with an LLM in LLaVA style. After the results were added, the reviewers  acknowledged the effectiveness of ToVE with LLM and improved their scores.

---

### Decision · Program_Chairs · 2025-01-22

Accept (Poster)